# Exploring the Molecular Aspects of Myeloproliferative Neoplasms Associated with Unusual Site Vein Thrombosis: Review of the Literature and Latest Insights

**DOI:** 10.3390/ijms25031524

**Published:** 2024-01-26

**Authors:** Erika Morsia, Elena Torre, Francesco Martini, Sonia Morè, Antonella Poloni, Attilio Olivieri, Serena Rupoli

**Affiliations:** 1Hematology Clinic, Azienda Ospedaliero Universitaria delle Marche, 60126 Ancona, Italy; 2Department of Clinical and Molecular Sciences (DISCLIMO), Università Politecnica delle Marche, 60126 Ancona, Italy; 3Clinic of Gastroenterology, Hepatology, and Emergency Digestive Endoscopy, Azienda Ospedaliero Universitaria delle Marche, 60126 Ancona, Italy

**Keywords:** myeloproliferative neoplasms, splanchnic vein thrombosis, cerebral vein thrombosis, clonal hematopoiesis of indeterminate potential

## Abstract

Myeloproliferative neoplasms (MPNs) are the leading causes of unusual site thrombosis, affecting nearly 40% of individuals with conditions like Budd–Chiari syndrome or portal vein thrombosis. Diagnosing MPNs in these cases is challenging because common indicators, such as spleen enlargement and elevated blood cell counts, can be obscured by portal hypertension or bleeding issues. Recent advancements in diagnostic tools have enhanced the accuracy of MPN diagnosis and classification. While bone marrow biopsies remain significant diagnostic criteria, molecular markers now play a pivotal role in both diagnosis and prognosis assessment. Hence, it is essential to initiate the diagnostic process for splanchnic vein thrombosis with a *JAK2 V617F* mutation screening, but a comprehensive approach is necessary. A multidisciplinary strategy is vital to accurately determine the specific subtype of MPNs, recommend additional tests, and propose the most effective treatment plan. Establishing specialized care pathways for patients with splanchnic vein thrombosis and underlying MPNs is crucial to tailor management approaches that reduce the risk of hematological outcomes and hepatic complications.

## 1. Introduction

### 1.1. MPNs and Thrombotic Risk

Thrombosis remains a major problem in patients with myeloproliferative neoplasms (MPNs). Indeed, this group of diseases, which include polycythemia vera (PV), essential thrombocythemia (ET), and myelofibrosis (MF), is characterized by the uncontrolled clonal expansion of multipotent bone marrow progenitors driven by acquired mutations in the *JAK2*, *CALR*, and *MPL* genes [1]. This expansion of the mutated clone initiates an inflammatory response that influences the development of related vascular complications in addition to disease progression [2].

The prospective analysis of thrombotic events in the European collaboration on low-dose aspirin study (ECLAP) found that cardiovascular events accounted for 45% of all-cause mortality in PV at 2.7 years follow-up, compared with hematologic transformation in 13% [3,4]. The risk of thrombosis appears to be highest at the time of initial diagnosis [5]. The rate of arterial and venous thrombosis in MPN patients has been estimated to be increased by 3-fold and 10-fold, respectively, compared with the general population [5]. In prospective studies involving ET, the combined rates of fatal and nonfatal thrombotic events varied from 0.9 to 2.6 per 100 people–years. Notably, the incidence of arterial events was 2–3 times higher compared to that of venous events [6]. In a recent study that involved 642 MF patients and 2568 matched controls, it was found that MF was independently associated with an increased risk of venous thromboembolism (VTE), but not with arterial thromboembolism. Among VTE, unusual site thrombosis occurred almost exclusively in patients with myelofibrosis, with four events of Budd–Chiari syndrome (BCS) compared to none in the control group, and two mesenteric vein thrombosis events compared to one in the control group. These events were also more likely to occur around the time of MF diagnosis [7].

In recent years, new insights into the pathogenesis of arterial and venous thrombosis associated with MPNs and the intricate interplay among blood cells, the endothelium, and the hemostatic system have emerged [8]. In the context of unusual site thrombosis associated with MPN, the available information regarding the pathogenesis is limited.

### 1.2. MPN and Unusual Site Thrombosis

Venous thromboses in unusual sites are rare and heterogenous manifestations of VTE and include cerebral vein thrombosis (CVT), retinal vein obstruction (RVO), and splanchnic vein thrombosis (SVT), which encompasses portal vein thrombosis (PVT), thrombosis of the hepatic veins causing BCS, mesenteric veins, and splenic veins. MPNs are the most frequent cause of PVT and BCS, found in about 30–40% of patients. In addition, inherited thrombophilia is present in at least one-third of patients with SVT. However, diagnosing inherited deficiencies of antithrombin, protein C, and protein S becomes challenging when liver impairment is present, leading to a reduced synthesis of natural anticoagulant. This challenge does not apply to the factor V Leiden and prothrombin G20210A mutations. Prothrombin G20210A mutation has consistently shown a high prevalence in various series of patients with extrahepatic portal vein obstruction (EHPVO), while factor V Leiden seems to be more prevalent in patients with BCS [9]. A meta-analysis indicates a 4.5-fold increased risk for EHPVO in prothrombin G20210A carriers and a twofold increased risk in factor V Leiden carriers [10].

All types of MPNs have been reported to cause SVT [11,12]. MPN-associated SVT (MPN-SVT) is a rare and often life-threatening condition, with mortality estimated at 40% at 5 years [13]. However, the clinical course is heterogeneous and, consequently, the impact of the thrombotic event on survival remains debated.

The prevalence of SVT in MPNS is estimated as 10–13% and it frequently serves as an initial manifestation of MPN [12,14]. CVT complicated MPNs in 1% of cases and nearly 50% of patients have a concomitant diagnosis of the two diseases [15].

Given its rarity and the paucity of robust studies, SVT and CVT management has often been extrapolated from experience of lower-limb DVT and pulmonary embolism (PE), and remains a challenging entity to manage clinically due to the high risk of recurrence and bleeding events [16].

The pathogenesis of unusual site thrombosis is still under study, as well as the identification of clinical risk factors and natural history, highlighting the need to expand the available data.

### 1.3. MPNs: Driver and Additional Mutations

MPNs are primarily initiated and promoted by acquired mutations in one of three pivotal driver genes: *JAK2*, *CALR* and *MPL*. These mutations can independently trigger and advance MPNs, without necessitating additional cooperative mutations. *JAK2 V617F* is prevalent in over 95% of PV cases and is also found in approximately half of ET or PMF patients. *CALR* and *MPL* mutations are also responsible for the onset of ET and PMF [17]. In roughly 10% of MPN cases, often referred to as “triple negative”, none of the known driver gene mutations can be identified. However, the common thread among all three driver gene mutations and triple-negative MPN is the constitutive activation of the Janus kinase-signal transducer and activator of transcription (JAK/STAT) signaling pathway [18,19].

JAK2-activated blood cells not only lead to an increase in cell count, which results in higher blood viscosity, but also bring about intrinsic changes in these cells. In translational studies involving MPN patients, it has been observed that leukocytes release procoagulant substances, platelets tend to spontaneously aggregate, and red blood cells exhibit enhanced adhesion to the vascular endothelium [20,21,22]. Additionally, *JAK2 V617F* is not restricted to hematopoietic cells; it is also likely to be present in the endothelial cells of the liver and spleen, thereby promoting the expression of endothelial P-selectin and contributing to thrombosis [23,24]. However, the expression of JAK2 in endothelial cells within the liver has yet to be definitively confirmed [25].

Furthermore, the mechanisms behind heightened vascular events and bone marrow-spleen neoangiogenesis remain elusive. Recently, some authors explored *JAK2* MPN driver mutations in endothelial progenitor cells (EPCs) or mature endothelial cells (ECs), finding that 70% of PMF patients shared mutations between hematopoietic stem/progenitor cells (HSPCs) and circulating ECs (CECs). CECs harbored myeloid-associated mutations, including *JAK2 V617F*. These results suggest a primary involvement of ECs in MPN, possibly originating from a common precursor cell. However, the acquisition of myeloid-associated mutations by ECs remains an open question [26].

Recent findings among patients with MPNs have unveiled the concurrent presence of somatic mutations linked to epigenetic regulation, messenger RNA splicing, transcriptional mechanisms, signal transduction, and DNA repair processes. The integration of genetic data into clinical practice is already enhancing patient care by offering improved disease monitoring and valuable prognostic insights into disease progression. Although current therapeutic approaches for MPNs have limitations in terms of their ability to modify the disease, the expanding knowledge of the genetic underpinnings of MPNs is presenting new potential avenues for targeted treatments [27,28,29].

## 2. Myeloproliferative Neoplasms Associated with Unusual Site Thrombosis

### 2.1. Peculiar Clinical Presentation

MPN-SVT represents a distinctive manifestation of MPNs characterized by specific disease features. Most individuals affected by MPN-SVT are young women and predominantly diagnosed with PV, with a low *JAK2V617F* VAF [30,31,32]. While these unique characteristics of MPN-SVT have been documented previously, the underlying mechanisms responsible for thromboses in these MPN patients remain unknown. The elevated risk of SVT in MPNs could be attributed to the distinct regulation of hemostasis in specific vascular regions. Endothelial cells play a pivotal role in modulating thrombosis and can express both pro- and anticoagulant factors contingent on their local environment and cell type [33]. A possible explanation of the interaction between portal flow characteristics and the influence of the JAK2 mutation will be provided in the following paragraph. This notion aligns with the observation that MPNs tend to manifest in specific anatomical locations within the splanchnic venous system. For instance, in MPN patients with BCS, thrombosis predominantly occurs in the large hepatic veins, while BCS patients with other hypercoagulable disorders typically experience thrombosis in the inferior vena cava [34]. Unique signaling pathways within the splanchnic venous system can lead to distinctive interactions with mutated blood cells. In conditions like paroxysmal nocturnal hemoglobinuria (PNH), which involves complement activation and a heightened risk of SVT, the presence of gut-derived antigens in splanchnic veins can trigger localized complement activation, resulting in increased rates of hemolysis and thrombotic events [35].

Moreover, the clinical features often present as a latent condition, which can pose challenges in diagnosing MPNs in the context of SVT. The primary reason for the absence of elevated blood counts in these cases is frequently linked to the presence of portal hypertension and splenomegaly arising from SVT and MPN. This situation can lead to the sequestration of blood cells and hemodilution, effectively concealing the typical clinical features of MPNs. Researchers have observed that “masked” patients often have lower hemoglobin levels but exhibit increased plasma volumes due to splenomegaly or the formation of SVT [36]. These effects can complicate the diagnosis of MPNs by altering some of the crucial diagnostic criteria, such as elevated peripheral blood cell counts or splenomegaly. In such situations, bone marrow biopsy has long been considered the most dependable diagnostic tool for identifying MPNs [12,37,38]. Further research has demonstrated that incorporating genetic mutation screening significantly enhances the precision of diagnosing MPNs within the context of unusual-site thrombosis. As a result, it is strongly recommended to integrate genetic testing for *JAK2* mutations into standard clinical protocols for investigating the underlying factors contributing to SVT in patients [39].

### 2.2. Role of Driver Mutation in Diagnosis and Prognosis

The *JAK2 V617F* mutation is the most common genetic alteration detected in individuals with SVT and an associated MPN, as previously mentioned [40]. Additionally, a substantial proportion of patients with SVT, including approximately 40% of BCS patients and 28% of those with PVT, are found to be positive for the *JAK2 V617F* mutation [41]. Despite the considerable portion of patients with splanchnic venous thrombosis, a small subset of those CVTs can be identified as carriers of the *JAK2 V617F* mutation, even when they do not display evident signs of MPNs. Specifically, the *JAK2 V617F* mutation was detected in 4.8% of patients with CVT who did not have a clear manifestation of MPN [42,43].

Therefore, it is advisable to include *JAK2* testing as part of the initial assessment for MPNs in any SVT patient. This approach has unveiled that a noteworthy percentage of SVT individuals (about 15–17%) test positive for the *JAK2 V617F* mutation, even if they do not meet the diagnostic criteria outlined by the World Health Organization (WHO) for an MPN [41]. In fact, one study suggests that JAK2 testing can identify an additional 30% of MPN patients who do not display clear clinical signs of the disease [44,45]. It is plausible that individuals with SVT who carry the *JAK2 V617F* mutation but do not exhibit obvious clinical manifestations of myeloproliferative neoplasms may represent a unique MPN subtype or an early stage of the disease, and these SVTs may manifest during the early phase of MPNs, preceding the development of overt clinical characteristics. Indeed, a meta-analysis revealed that more than 50% of *JAK2*-mutated patients who did not have an MPN diagnosis when they experienced SVT later developed an MPN during follow-up [46]. Moreover, the hypothesis that SVTs represent early signs of MPN might partly explain why MPN-SVT patients tend to be younger and exhibit lower *JAK2 V617F* allele burdens [47].

Moreover, the *JAK2 V617F* variant allele frequency (VAF) is a key determinant of outcomes in MPN, including thrombosis. The data have shown that *JAK2 VAF* in PV positively correlates with circulating CD34+ cell levels, bone marrow cellularity, and splenomegaly [48,49,50]. In addition, a VAF of ≥50% increases the risk of VTE, even after adjusting for previous events, white blood cell count, and age [50,51,52]. This association was observed in both conventionally low-risk and high-risk PV patients [50]. Moreover, JAK2 V617F allele burden >50% is associated with an adverse MPN-SVT hematologic outcome (incidence of myelofibrosis, blast phase MPN or death) [53]. The role of the temporal trend in the VAF of *JAK2 V617F* remains unclear, and an MRD assessment warrants further evaluation. Recently, digital PCR has demonstrated suitable sensitivity and specificity for MRD monitoring in *JAK2 V617F* patients and *CALR*-mutated patients [54,55].

The effect of *JAK2 V617F* VAF on both the myeloproliferation and thrombotic risks observed in PV is mirrored in studies of clonal hematopoiesis (CH) among non-MPN populations. Individuals with *JAK2 V617F*-positive CH exhibit significantly elevated white blood cell counts, platelet counts, and hemoglobin levels. Moreover, they experience higher rates of arterial and venous thrombotic events compared to those without CH [56,57,58,59]. Cordua and colleagues explored the relationship between *JAK2 V617F* CH and VAF in a cohort of 19,958 participants from the Danish General Population, employing an assay capable of detecting VAF as low as 0.009%. They identified *JAK2 V617F*-positive CH in 3.1% of participants, with a median VAF of 2.1%. Notably, individuals with *JAK2 V617F* CH and higher VAFs exhibited elevated blood cell counts and had a greater susceptibility to venous thrombosis, PE, or cerebrovascular events [60]. However, despite these associations, another study observed that young females tended to exhibit a lower burden of the *JAK2* mutant allele, even though they are at a higher risk of developing SVTs [61]. These findings may suggest an ‘all or nothing’ effect of the *JAK2* mutation, which could potentially be influenced by additional clinical risk factors like concurrent hypercoagulable disorders.

In conclusion, the *JAK2 V617F* mutation could potentially exert localized effects on the splanchnic venous system. This vascular system is distinctive in that it features a much slower blood flow, leading to prolonged interactions between blood and sinusoidal endothelial cells. In laboratory models, it has been demonstrated that red blood cells in PV show increased adhesion to endothelial laminin. This adhesiveness is particularly pronounced at low shear rates [62]. Moreover, the portal venous system receives inputs from the intestines, altering its immunogenicity. This renders the vessels more susceptible to activated platelets and conditions of high viscosity, such as elevated cell counts. These distinctive characteristics may render the splanchnic venous system particularly vulnerable to perturbations induced by *JAK2 V617F*. The precise mechanism through which changes in the endothelium contribute to unusual site thrombosis remains uncertain and requires additional research. Considering the distinct nature of the splanchnic venous system, it is probable that the pathways leading to SVT formation differ from arterial and DVT pathways (as shown in Figure 1).

### 2.3. Comprehensive Genomic Profiling of Thrombosis Risk

#### 2.3.1. Other Molecular Drivers of MPN-Associated SVT

While the *JAK2 V617F* mutation plays a pivotal role in MPN associated with SVT, it is noteworthy that around 14–20% of SVT patients with MPN lack the *JAK2* mutation [11,41]. From a clinical risk perspective, *JAK2*-unmutated MPN-SVT patients do not exhibit significant statistical differences compared to their *JAK2*-mutated controls [30]. This raises the question of whether other mutations may contribute to the prothrombotic state. Specifically, a range of mutations with an increased incidence in MPNs have been identified. Only recently have researchers started exploring the role of these mutations in SVT. There is a very low prevalence of *CALR* mutations in general patients with SVTs (0–4.88%). In a recent meta-analysis, the pooled proportion of *CALR* mutations in MPN-SVT *JAK2 V617F*-negative patients was 15.16% [63].

Following the identification of the *JAK2 V617F*, *MPL* and *CALR* mutations, numerous non-driver somatic mutations affecting genes responsible for various cellular processes have been identified in MPN. These numerous non-driver somatic mutations affecting genes involved in the regulation of various cellular processes, such as epigenetics, transcription, mRNA splicing, or signaling pathways, have been uncovered in MPNs [27]. These mutations may coexist with or occur independently of the three driver mutations. Sometimes, they can serve as indicators of clonality, which is a significant diagnostic criterion for MPNs. However, it is crucial to highlight that the presence of these mutations alone cannot lead to an MPN diagnosis. Furthermore, many of these mutations have been found in otherwise healthy individuals, and their prevalence increases with age. This phenomenon is known as the clonal hematopoiesis of indeterminate potential (CHIP), and its precise clinical implications are still being investigated. CHIP is considered a myeloid precursor lesion according to the WHO 2022 classification. Indeed, WHO 2022 defines CHIP as the detection of one or more somatic mutations with VAF ≥ 2% involving selected genes, with an absence of unexplained cytopenias and an absence of diagnostic criteria for defined myeloid neoplasms [64].

Actually, in the context of CHIP, mutations in *DNMT3A*, *TET2*, and *ASXL1* have already been shown to carry a 1.9-fold higher risk of coronary disease and a four-fold higher risk of myocardial infarction [57,65,66]. Interestingly, recent reports indicate that *TET2* and *DNMT3A* genes are both associated with an elevated thrombotic risk in PV [67]. A recent link between CHIP and VTE risk has emerged. This connection was initially explored in a study investigating whether individuals with *JAK2 V617F*-positive CHIP had a population of clonal neutrophils predisposed to producing neutrophil extracellular traps associated with VTE [68]. In a substantial case-control group of 10,893 individuals, this study revealed a strong association between *JAK2 V617F*-mutant CHIP and significant VTE events [69]. However, subsequent research yielded conflicting findings. In a study of 11,695 patients with solid cancers, no significant relationship between any CHIP mutations, including *JAK2 V617F*, and VTE risk was observed [70]. Nevertheless, a preliminary retrospective observational study of 61 subjects with unprovoked PE found that 20% had somatic mutations associated with CHIP [70].

Moreover, recent discoveries disclose a higher prevalence of *JAK2 V617F*-positive CHIP in precapillary pulmonary hypertension (PH) patients compared to healthy subjects. Mice models showed that JAK2 V617F-induced STAT3 phosphorylation further upregulates ALK1-Smad1/5/8, inducing PH and arterial remodeling in mice [71,72].

Few studies tried to clarify the influence of driver and other mutations on both arterial and venous thrombosis in MPN. A recent study on ET patients showed the prognostic interplay between extreme thrombocytosis and *CALR* mutations in influencing arterial event incidence at diagnosis. Notably, the favorable impact of *ASXL1/RUNX1/EZH2* mutations on arterial thrombosis was apparent, both before and after diagnosis. This suggests that the infrequent occurrence of high-risk mutations may serve as a marker for a distinct disease phenotype, possibly related to occult prefibrotic myelofibrosis [73,74].

A recent study confirmed that the order of mutations has relevant implications in MPN-SVT patients, since cases where *JAK2* mutations occurred first have already been demonstrated to be typically associated with an increased risk of thrombosis [75]. Moreover, *TET2* was also the most frequently mutated gene in MPN-SVT, occurring in 51.7% of patients, followed by *DNMT3A,* occurring in 31.0% [76,77]. Notably, some of these genes have critical functions beyond their enzymatic roles, with *TET2* emerging as a significant regulator of hematopoietic stem cell biology [78]. *TET2* mutations have been observed in a substantial portion of both *JAK2 V617F*-positive and *JAK2 V617F*-negative MPN patients, suggesting their involvement as early events in disease development [79]. Furthermore, *TET2* deficiency has been linked to an increased pro-inflammatory phenotype in murine macrophages, potentially promoting the development of atherosclerosis and thrombosis [80]. Next-generation sequencing (NGS) profiling in MPN patients with SVT could identify certain mutations associated with poorer event-free and overall survival (Table 1). Recent retrospective data showed that the presence of chromatin/spliceosome/*TP53* mutations in MPN-SVT is associated with adverse outcomes [53]. This underscores the potential of a patient’s molecular profile to provide critical prognosis information and guide therapeutic decisions. CHIP mutations, commonly found in genes like *TET2*, *DNMT3A* and *ASXL1*, are frequently associated with an increased risk of venous thromboembolic events, including SVT. These mutations contribute to a prothrombotic state, with the understanding that they are often considered “founder mutations”, present in both CHIP and MPNs, suggesting a shared clonal origin. Although MPNs diagnosed at the time of an SVT are considered to be in the early stages of their disease onset, recent genomic findings reveal a complex genetic landscape, further endorsing the recent perspective that the preclinical evolution of these malignancies dates back to well before the time of diagnosis [81].

#### 2.3.2. Next-Generation Sequencing in the Diagnosis of Non-Cirrhotic Splanchnic Vein Thrombosis

Incorporating non-driver mutations into the diagnostic process may aid in the identification of MPNs in patients with SVT. Recent studies have explored the potential of NGS to enhance the diagnosis of SVT. For example, Magaz and colleagues reported that NGS could detect genetic variants in non-driver genes in 37.8% of patients with idiopathic SVT, with several cases exhibiting triple-negative MPNs [82]. This study raises intriguing insights, indicating that a substantial portion of idiopathic, not cirrhotic, SVT patients may harbor CHIP. As previously mentioned, emerging research has demonstrated the functional impacts of CHIP mutations, particularly *DNMT3A*, *TET2*, *ASXL1* and *JAK2*, on platelets, which aligns with the investigators’ findings and a susceptibility to SVT. While this discovery does not directly alter the anticoagulation strategies for SVT, it highlights the need for vigilant hematologic monitoring and sets the stage for potential clinical trials to assess chemoprophylaxis in patients with CHIP and not-cirrhotic PVT. Moreover, Carrà et al. present findings from 15 consecutive cases of idiopathic SVT that exhibited mutations in one or more of the thirty myeloid genes analyzed using NGS. In seven of these cases, the authors detected CHIP, characterized by the acquisition of somatic mutations in leukemia-associated driver genes in individuals lacking underlying hematologic malignancies, including *DMT3A.* Notably, the latter study, with a VAF threshold set at >2%, reported a CHIP prevalence of 46%, which is the highest ever reported, nearly 10-fold higher a CHIP prevalence than expected in the general population of similar age [57,83]. Although these results are based on a limited number of patients, they strongly suggest a potential role of CHIP in idiopathic SVT.

## 3. Discussion

Managing patients with SVT and diagnosing MPN is often challenging for specialists, including hematologists and gastroenterologists. The diagnosis of an SVT-correlated MPN is often complicated by the latent nature of myeloproliferation. Additionally, the effectiveness of cytoreductive treatments remains unverified, introducing uncertainty in therapeutic approaches. The recent Baveno VII conference, titled “Personalized Care for Portal Hypertension”, brought together experts in hematology, hepatology, and internal medicine to consolidate the latest knowledge on the pathophysiology and treatment of portal hypertension in patients [84]. Regarding individuals with idiopathic SVT, the conference highlighted specific recommendations for initiating the diagnostic evaluation of an underlying MPN, starting with screening for the *JAK2 V617F* mutation in the peripheral blood. In cases where the *JAK2 V617F* mutation is not detected, further genetic investigations for MPN should be considered, including other driver mutations, such as *CALR* mutations, as well as conducting a comprehensive screening for other MPN-associated mutations using NGS [85]. Lastly, for patients without any MPN biomarkers and no other identified cause of SVT, the possibility of a bone marrow biopsy should still be considered, regardless of peripheral blood cell counts. This is crucial for formally ruling out an underlying MPN, particularly in cases of BCS, due to the potential for a worse long-term prognosis and a higher risk of severe complications if an underlying MPN is left undiagnosed. (Figure 2).

The idea that SVT patients might represent an earlier phase of MPN carries significant implications when examining the molecular basis of MPN development. There is a prevailing hypothesis that hematologic cancers emerge through a series of genetic mutations, occurring step by step. These mutations provide survival and growth advantages to hematopoietic stem cells, gradually leading to clonal hematopoiesis and ultimately culminating in the onset of malignancy. This concept suggests that SVT, which may precede the classic clinical signs of MPNs, could serve as an early marker of the genetic alterations and clonal evolution that underlie MPN pathogenesis [39]. Although initial mutations can facilitate the expansion of a clone of cells, the subsequent cooperating mutations that occur later in the process drive the development of hematologic cancers. In the case of *JAK2*-mutated patients who do not display the clinical signs and symptoms of an MPN, it is conceivable that they lack the cooperating mutations necessary for the uncontrolled proliferation of blood cells. For example, the *JAK2 V617F* mutation has been detected in elderly individuals who do not exhibit clear signs of an MPN, but who have increased risk of developing a subsequent hematologic malignancy and cardiovascular mortality [65,86]. The transition to an evident MPN state, characterized by increased cell counts, might only become evident when additional mutations are acquired. Hence, patients with a confirmed *JAK2* mutation may be regarded as being in a “pre-malignant” stage, constituting a distinct group that offers valuable insights into the clonal evolution of MPNs.

The use of NGS data can prevent the underdiagnosis of MPNs and guide the selection of optimal therapeutic strategies for MPN-SVT. It is becoming increasingly clear that evaluating the complete molecular profile of SVT patients using targeted NGS at the time of diagnosis is of pivotal importance, particularly for younger patients and those with severe forms of SVT. Achieving an early diagnosis of MPN in these cases allows for the use of cytoreduction strategies. On the other hand, a pooled analysis of 1500 cases of MPN patients showed that cytoreduction therapy by hydroxyurea could prevent arterial and late-venous thrombotic recurrences but fails in the splanchnic venous district [87]. The reasons behind this finding are challenging to elucidate. One possible speculation is that patients with SVT tend to have less frequent hypercythemia, reducing the criticality of cytoreduction in this context. Moreover, considering the distinctive nature of the splanchnic venous system, it is probable that the mechanisms driving SVT formation differ from those causing arterial and VTE events at more common sites. Therefore, when managing SVT, which can occur as the initial sign of MPN or as a complication during the disease course, deciding on cytoreduction usage becomes a key clinical decision.

While the detection of non-driver somatic mutations can signify clonal hematopoiesis, it is essential to consider their significance in diagnosing MPNs cautiously, as a considerable number of healthy individuals may harbor CHIP. Although the exact mechanisms by which CHIP contributes to SVT formation are not fully elucidated, it is believed that these mutations disrupt the balance of clotting factors, elevating the risk of thrombus formation. Furthermore, they may influence platelet function, amplifying the overall hypercoagulable state. Additionally, the presence of additional mutations in MPN patients has been associated with specific clinical implications, including an elevated risk of thrombotic events, disease progression, and unfavorable outcomes compared to those without mutations [53]. Understanding the intricate relationships between these mutations and SVT formation and their clinical consequences is essential for enhancing the diagnosis and management of MPNs in patients with SVT. The identification by NGS of CHIP in SVT patients could help to explain SVT in younger individuals without conventional risk factors, emphasizing the need for comprehensive evaluations and preventive measures.

The presence of CHIP also could impact long-term management, as these patients may be at higher risk for other hematologic malignancies and cardiovascular diseases [88]. Regular monitoring and follow-up are vital to detect potential disease progression. Lifestyle changes and preventive measures, like long-term anticoagulation therapy, may be considered to reduce the risk of thrombotic events in SVT patients with CHIP.

The studies conducted by Carrà et al. and Magaz et al. offer valuable insights into the role of CHIP as a risk factor for VTE and unusual-site VTE. However, several questions remain unanswered, necessitating further investigation through large multicenter studies. One crucial aspect that requires clarification is whether the risk of VTE is associated with mutations in specific genes within the CHIP framework. Furthermore, it is essential to determine whether patients with CHIP-associated mutations face an elevated risk of VTE recurrence. Approximately 43% of patients with CHIP exhibited JAK2 V617F mutations, alongside DNMT3A, and EZH2 mutations. These findings raise questions regarding the differentiation between MPN-specific mutations and CHIP-associated ones. Steemsa et al. previously expressed similar concerns about CHIP in the general population, particularly related to individuals with cytopenia, but lacking other clinical indicators of myelodysplastic syndrome (MDS) [89]. Consequently, additional research is warranted to thoroughly investigate these concerns and offer a clearer understanding of the clinical implications surrounding CHIP and its associated mutations within the context of SVT.

## 4. Conclusions and Future Directions

There are still some aspects that have yet to be clarified in the context of MPN-SVT. Firstly, robust data are lacking to consider SVT as an early presentation of subsequent MPN, with associated therapeutic implications such as the early use of cytoreductive therapy. Secondly, it is of the utmost importance to ascertain whether the VAF in *JAK2 V617F* plays a role in MPN-SVT, not only in diagnosis and prognosis, but also in achieving thrombosis-related outcomes (e.g., recanalization, recurrence). In addition to *JAK2*, additional mutations in MPNs have a clear prognostic role, but their role in SVT pathogenesis is unclear, and there is no strong consensus among researchers regarding the use of non-*JAK2*-mutation screening in SVT patients. By incorporating NGS data with clinical parameters, novel prognostic tools could be developed, enabling the identification of MPN-SVT patients at risk of experiencing a dismal long-term outcome. For high-risk patients, considering a disease-altering treatment might be advisable to mitigate the potential for hematological transformation when feasible. Conversely, low-risk patients displayed no disease progression over a decade, and, when necessary, cytoreductive therapy with hydroxyurea, in addition to anticoagulation, appears suitable for diminishing the risk of thrombosis recurrence.

Furthermore, a complex relationship exists between CHIP, MPN, and unusual-site thrombosis. CHIP mutations often co-occur with MPNs and play a role in MPN development and thrombotic risk. MPNs are inherently linked to an elevated risk of SVT. Detecting CHIP mutations in individuals with MPNs and SVT carries important implications for risk evaluation, offering insights into therapeutic choices and long-term care planning. These interconnections emphasize the need for a comprehensive understanding of the molecular and clinical aspects of these conditions to optimize patient management and risk assessment.

## Figures and Tables

**Figure 1 ijms-25-01524-f001:**
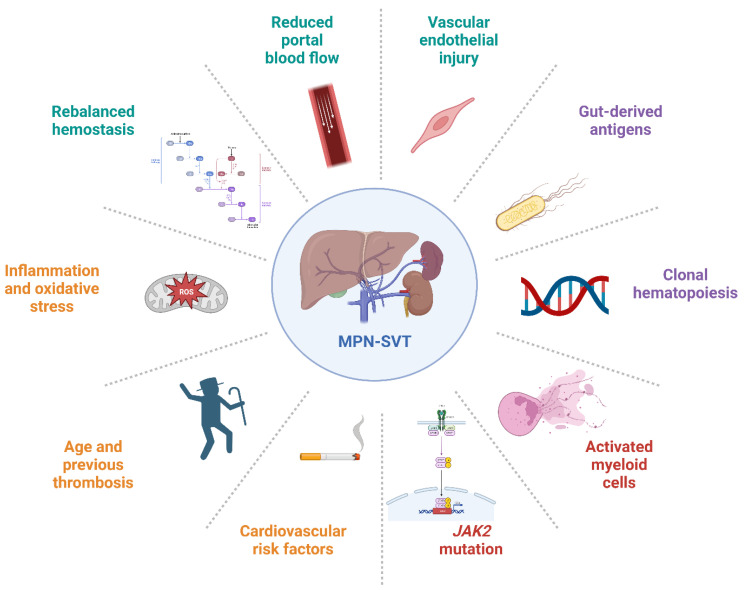
Clinical, molecular, and emerging factors that play a pivotal role in the pathogenesis of MPNs associated with SVT. In detail, risk factors related to MPN are highlighted in red; risk factors associated with patient characteristics are indicated in orange; portal-flow-related risk factors are presented in blue; and emerging factors are depicted in purple.

**Figure 2 ijms-25-01524-f002:**
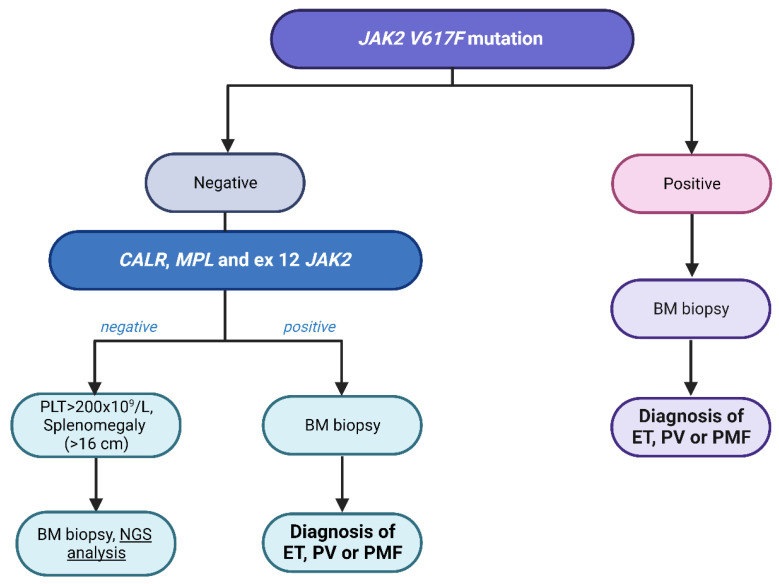
Diagnostic evaluation for myeloproliferative neoplasms (MPNs) associated with splanchnic vein thrombosis (SVT). BM: bone marrow; CALR: calreticulin, ET: essential thrombocythemia, JAK2: janus kinase 2; MPL: myeloproliferative leukemia virus oncogene; NGS: next-generation sequencing, PLT: platelets; PMF: primary myelofibrosis; PV: polycythemia vera.

**Table 1 ijms-25-01524-t001:** Mutational landscape in MPN-SVT.

Corresponding Effect	Gene Mutated	Rate (%)	References
Driver mutation	*JAK2 V617F* VAF > 50%	21–22	[53,54,55,56,57,58,59,60,61,62,63,64,65,66,67,68,69,70,71,72,73,74,75,76]
	*JAK2 V617F* VAF < 50%	74–62	[53,54,55,56,57,58,59,60,61,62,63,64,65,66,67,68,69,70,71,72,73,74,75,76]
	*CALR*	5	[53,54,55,56,57,58,59,60,61,62,63,64,65,66,67,68,69,70,71,72,73,74,75,76]
	*MPL*	7	[76]
DNA methylation	*TET2*	21–28	[53,54,55,56,57,58,59,60,61,62,63,64,65,66,67,68,69,70,71,72,73,74,75,76]
	*DNMT3A*	11–17	[53,54,55,56,57,58,59,60,61,62,63,64,65,66,67,68,69,70,71,72,73,74,75,76]
	*IDH1-IDH2*	6	[53]
Chromatin Spliceosome	*ASXL1*	8–11	[53,54,55,56,57,58,59,60,61,62,63,64,65,66,67,68,69,70,71,72,73,74,75,76]
	*EZH2*	2–3	[53,54,55,56,57,58,59,60,61,62,63,64,65,66,67,68,69,70,71,72,73,74,75,76]
	*SF3B1*	3	[53]
	*SRSF2*	1	[53]
	*U2AF1*	1–4	[53,54,55,56,57,58,59,60,61,62,63,64,65,66,67,68,69,70,71,72,73,74,75,76]
	*ZRSF2*	1	[53]
Other	*TP53*	4	[53,54,55,56,57,58,59,60,61,62,63,64,65,66,67,68,69,70,71,72,73,74,75,76]

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
