# Peer review of "Exploring the Molecular Aspects of Myeloproliferative Neoplasms Associated with Unusual Site Vein Thrombosis: Review of the Literature and Latest Insights"

_ijms, 2024, doi:10.3390/ijms25031524_

Round 1

Reviewer 1 Report

Comments and Suggestions for Authors

The article entitled “Exploring the molecular aspects of myeloproliferative neoplasms associated with unusual site vein thrombosis: review of literature and latest insights” review the results of the scientific literature about the splanchnic vein thrombosis in myeloproliferative neoplasms and talks about the myeloproliferative molecular genetic as cause of a rare and poorly frequent venous thrombotic  complication. The structure of this article is well-designed and the informations are exhaustive. I have only an observation regarding the aspects of inherited or acquired thrombophilic factors (Factor V Leiden and G20110 A mutations, LAC) in splanchnic vein thrombosis. Therefore, I suggest to very briefly discuss this issue. In addition, I suggest to add the reference regarding CHIP and thrombosis “Barbui T et al Blood Advances 2023”. I think that this article is suitable for publication in according to reviewer’s suggestions

Author Response

I am pleased to read the review of my article titled “Exploring the molecular aspects of myeloproliferative neoplasms associated with unusual site vein thrombosis: review of literature and latest insights” and I have incorporated all the suggested changes, in particolar those regarding inherited or acquired thrombophilic factors. I hope that the article has now improved and is suitable for publication.

Erika

Reviewer 2 Report

Comments and Suggestions for Authors

In the manuscript entitled “Exploring the molecular aspects of myeloproliferative neoplasms associated with unusual site vein thrombosis: review of literature and latest insights”, Morsia et al. reviewed the venous thrombosis associated with MPNs and JAK2-mutated clonal hematopoiesis. The authors emphasize the role of testing JAK2V617F and other MPN drivers in the diagnosis of venous thrombosis in unusual site, which is an important aspect. There are some concerns in this manuscript described below.

1.     Line 40: “Risk of thrombosis appears highest at the time of initial diagnosis.” is correct statement, with no citation. Please cite the study that clearly showed this fact: Hultcrantz M, et al. Risk for Arterial and Venous Thrombosis in patients with myeloproliferative neoplasms: a population-based cohort study. Ann Intern Med. 2018;168:317-325.

2.     Budd-Chiari syndrome should be abbreviated on line 49, rather than line 60.

3.     Unusual sites for venous thrombosis are mentioned on lines 57-61. I would suggest adding an important thrombotic complication, chronic thromboembolic pulmonary hypertension (CTEPH), which is often associated with deep vein thrombosis in patients with MPNs and JAK2-mutated clonal hematopoiesis. Please cite following articles. 

(1)   Kimishima Y et al. Clonal hematopoiesis with JAK2V617F promotes pulmonary hypertension with ALK1 upregulation in lung neutrophils. Nat Commun. 2021;12(1):6177.

(2)   Montani D, et al. Clinical Phenotype and Outcomes of Pulmonary Hypertension Associated with Myeloproliferative Neoplasms: A Population-based Study. Am J Respir Crit Care Med. 2023;208(5):600-612.

4.     On line 70, PE should be described with full name.

5.     On lines 89 and 92, JAK2 should be “JAK2 V617F” or “JAK2 mutation”.

6.     On line 106, authors use “mutation allelic burden” whereas “VAF” is used in other sentences. Consistent use of “VAF” may be better. 

7.     The authors state clonal hematopoiesis (CH) in the paragraph from line 169 to line 184. Then, CHIP is described on line 224, far from line 184. The authors should remember that CHIP is a part of CH. In fact, latest WHO classification (Leukemia 2022;36:1703–1719) defines CHIP as a term referring specifically to CH harboring somatic mutations of myeloid malignancy-associated genes detected in the blood or bone marrow at a VAF of ≥ 2% in individuals without a diagnosed hematologic disorder.

8.     Line 288: variant allele frequency may be VAF because it is already described above.

9.     Line 387: VAF in JAK2 should be VAF in JAK2 V617F.

Comments on the Quality of English Language

There are some mistakes in this paper.

1.       Line 26 (keywords): potentia should be potential.

2.       Line 30: Is the word “contemporary” correct?

3.       Line 70: “MPN” may be “MPNs”.

4.       Line 108: unknow maybe unknown.

5.       Line 246: “Recent studies” may be “Recent study” because one paper is cited (ref67).

Author Response

I am pleased to read the review of my article titled “Exploring the molecular aspects of myeloproliferative neoplasms associated with unusual site vein thrombosis: review of literature and latest insights” and I have incorporated all the suggested changes you suggest.

I hope that the article has now improved and is suitable for publication.

Erika

Reviewer 3 Report

Comments and Suggestions for Authors

Morsia and colleagues presented a comprehensive revision of literature about the molecular aspects of MPNs characterized by unusual site vein thrombosis.

The review is well written and readable. I have some suggestions to improve the completeness of the manuscript.

- Line 89. The Authors reported that the gene JAK2 is present also in endothelial cells. The gene is present in every cells when no deletion occurs! If the Authors referred to the protein, JAK2 must be reported in capital letter, not italic. Conversely, if they referred to the gene, "expressed" is the most correct verb.

- Section 1.3. Recently, Farina et al (PMID 34685741) reported that JAK2 V617F mutation is frequently shared between hHSC and endothelial cells. In particular, they detected the mutation in circulating endothelial cells derived from blood vessels. Please, add a comment on these insights.

- Line 202. there is a space/typo error.

- Section 2.2. Digital PCR has been recently reported as a new tool to better quantify V617F+ alleles and more precisely assess the VAF (PMID 35741115, PMID 33973741). It is also able to detect very rare mutations within PBMC and/or BM cells (PMID 33261150). Please, add a brief comment on this new techniques and opportunity, also by referring to the suggested examples.

- Section 2.3.1. This part is full of info. I strongly suggest the authors to provide a table reporting the genes/genetic variants and the corresponding effects.

- Figure 1. A legend about the different colors used is needed.

I hope my suggestions will help the Authors in improving their manuscript.

Author Response

I am pleased to read the review of my article titled “Exploring the molecular aspects of myeloproliferative neoplasms associated with unusual site vein thrombosis: review of literature and latest insights” and I have incorporated all the suggested changes, in particolar those regarding ddPCR and endothelial cells.

I hope that the article has now improved and is suitable for publication.

Erika

Round 2

Reviewer 3 Report

Comments and Suggestions for Authors

The Authors satisfied all my requests and replied to all my questions.